# Automatic Identification for the Boundaries of InSAR Anomalous Deformation Areas Based on Semantic Segmentation Model

Yiwen Liang [1], Yi Zhang [2,*], Yuanxi Li [2] and Jiaqi Xiong [1]

1   Aerial Photogrammetry and Remote Sensing Bureau of China Administration of Coal Geology, Xi'an 710199, China; liangyw19@lzu.edu.cn (Y.L.); 201940100061@jxnu.edu.cn (J.X.)
2   Technology & Innovation Centre for Environmental Geology and Geohazards Prevention, School of Earth Sciences, Lanzhou University, Lanzhou 730000, China; liyq2021@lzu.edu.cn
*   Correspondence: zhangyigeo@lzu.edu.cn

**Abstract:** Interferometric synthetic aperture radar (InSAR) technology has become one of the mainstream techniques for active landslide identification over a large area. However, the method for interpreting anomalous deformation areas derived from InSAR data is still mainly manual delineation through human–computer interaction. This study focuses on using a deep learning semantic segmentation model to identify the boundaries of anomalous deformation areas automatically. We experimented with the delineation results based on an InSAR deformation map, hot spot map, and different combinations of topographic datasets to build the optimal model. The result indicates that the hot spot map, aspect, and Google Earth image as input features based on the U-Net model can achieve the best performance, with the precision, recall, F1 score, and intersection over union (IoU) being 0.822, 0.835, 0.823, and 0.705, respectively. Our method promotes the development of identifying active landslides using InSAR technology automatically and rapidly at a regional scale. Moreover, applying a new method for automatically and rapidly identifying potential landslides in susceptible areas is necessary for landslide hazard mitigation and risk management.

**Keywords:** landslide delineation; semantic segmentation; InSAR; deep learning





## 1. Introduction

Landslides are one of the most devastating natural disasters, causing huge economic losses and thousands of deaths every year around the world [1–3]. Affected by destructive earthquakes [4–6], rapid snowmelts [7–9], intense rainfalls [10–12], and human activities [13–16], landslides may tend to occur in susceptible areas, posing a great threat to local residents [17]. Rapidly obtaining the scopes of active potential landslides is helpful in order to take effective measures to save people's lives and property before disaster and provide useful information regarding landslide risk management for governments [3].

Surface deformation is an apparent indicator for identifying active potential landslides [18]. Although the traditional methods including the global positioning system (GPS), total stations, and inclinometers can detect surface deformation accurately, they are not suitable for large-scale detection [19]. Interferometric synthetic aperture radar (InSAR) technology can detect millimeter-level surface deformation over a large scale, with the characteristics of all-weather performance, a wide range, and high precision [20,21], and has been widely applied for potential landslide recognition [22–27]. The process of identifying active potential landslides using InSAR surface deformation is achieved through visual interpretation based on remote sensing images. The boundaries of active slopes can be determined by superimposing on optical remote sensing images with reference to surface geomorphological features (e.g., scarps, sliding masses, and bulging toes) [23], a process

which can delineate landslides accurately but is quite dependent on expert knowledge and time-consuming [28].

In order to identify the boundaries of anomalous InSAR deformation areas effectively and accurately, some researchers have proposed methods including spatial statistical analysis, object-based image analysis (OBIA), and deep learning models. Spatial statistical clustering analysis depends on the distribution of anomalous deformation points to obtain the areas with evident clustering properties in space [29,30], which is similar to the idea of delineating clustered anomalous deformation points in the manual interpretation. The identification results are consistent with the distribution of anomalous deformation points, but this process may result in omissions when landslides are partly reactivated. The OBIA method can segment separate objects with similar characteristics of texture, shape, and spatial structure [31–33], and then classifies them using a machine learning algorithm to produce the final classification results [34]. However, this method has high requirements regarding the segmentation scale and the used features, which are determined by means of visual judgment. No particular combination of segmentation scale and optimal classification features is suitable for all targets, implying that multiple experiments and practical experience are needed to obtain optimal results [35], limiting the applicability of the OBIA method. Deep learning semantic segmentation is a typical application in computer vision that aims to annotate every pixel within imagery with a specific semantic label [36–40] and has achieved remarkable performance in landslide delineation with innovative architectures [41–43]. Deep learning methods can easily obtain optimal parameters based on the back propagation strategy and multiple epochs for training [44], without the requirement for extracting image features. Although some research has pointed to identifying anomalous deformation areas, the corresponding work continues to only use InSAR deformation maps [42,45–47]. An appropriate combination of InSAR deformation maps, optical remote sensing images, and other supplementary data will be the future development trend, which is more akin to simulating the process of manual interpretation.

The upper reaches of the Yellow River, affected by the strong uplift of the Qinghai–Tibet Plateau, has become one of the regions where geological hazards are most severe in China [48]. Many ancient large-scale and unstable landslides are distributed on both sides of the Yellow River, which pose a great threat to local residents [49]. In addition, the construction and operation of hydropower stations may contribute to the development of unstable slopes in reservoir areas [49]. Identifying potential landslides is necessary for preventing the losses of life and property caused by landslides and ensuring the safe operation of hydropower stations. In addition, the sparse vegetation coverage of the area is favorable for InSAR analysis [50]. Therefore, the upper reaches of the Yellow River were selected as the study area to explore the method for identifying InSAR anomalous deformation areas rapidly and accurately.

In this paper, we use a combination of InSAR techniques, hot spot analysis, and a semantic segmentation model to establish identification models. We then compare the performance of different input features and different semantic segmentation models. Finally, we establish the optimal model and conclude its further directions. Our results contribute to establishing models for delineating active potential landslides automatically and provide valuable information for landslide susceptibility regions in terms of landslide early warnings and prevention over a large scale.

## 2. Study Area

The upper reaches of the Yellow River are located in the transitional zone between the Loess Plateau and the Qinghai–Tibetan Plateau [51]. Since the Late Pliocene, the Qinghai-Tibetan Plateau has experienced extensive uplift, while the upper reaches of the Yellow River and its tributaries have incised deeply into a series of intermountain basins and bedrock ranges [52], forming a landform with high reliefs and steep slopes, thus providing suitable topographic conditions for landslides' development [53]. A large number of historical landslides have developed along both sides of the upper reaches

of the Yellow River, and most landslides are distributed in Qunke–Jianzha Basin and Guide Basin [54]. Therefore, a 15 km buffer zone on both sides of the upper reaches of the Yellow River, from Guide Basin to Xunhua Basin, was selected as the study area (35°40′–36°20′N, 101°20′–102°0′E), in the east of Qinghai Province.

In the study area, the Yellow River flows from west to east through Guide Basin, Lijia Gorge, Qunke–Jianzha Basin, and Xunhua Basin. High valleys and flat intermountain basins, with altitudes varying from 1837 to 4421 m, dominate the topography of the study area (Figure 1b). Many ancient large-scale landslides are distributed in the study area including Xijitan landslide, Suozi landslide, Garang landslide, and so on [54–56], which are still unstable. The boundaries of these ancient large-scale landslides are shown in Figure 1. There are about nine faults developed in the study area, and most of these faults trend from NW to SE, with the Songba fault oriented from N to S. Tectonic movement is affected by the Xi Qinling–Jishishan fault located on the south side and the Lajishan fault located on the north side [49]. The lithology of the study area is mainly Triassic slate and fine-grained sandstone, Paleogene and Neogene sandstone and mudstone, and Quaternary sediments, characterized by heavy weathering (Figure 1c) [53]. The area is dominated by a continental semi-arid climate characterized by a dry and windy spring, a short and cool summer, a wet and rainy autumn, and a long and dry winter [50]. The average annual precipitation is 255–354 mm, which is mainly concentrated in summer and autumn, while the average annual evaporation is 1901–2136 mm [49]. The daily temperature ranges from −28 °C to 25 °C. The Lijia Gorge hydropower station was built in November 1999 with a normal storage water level of 2180 m a.s.l. The operation of a hydropower station may seriously affect the stability of slopes in reservoir areas due to the fluctuation of the water level of the reservoir [57]. In Lijia Gorge, several active landslides have developed, like the Lijia Gorge III landslide [49], which has seriously threatened the highway near the landslide and the operation of the hydropower station.

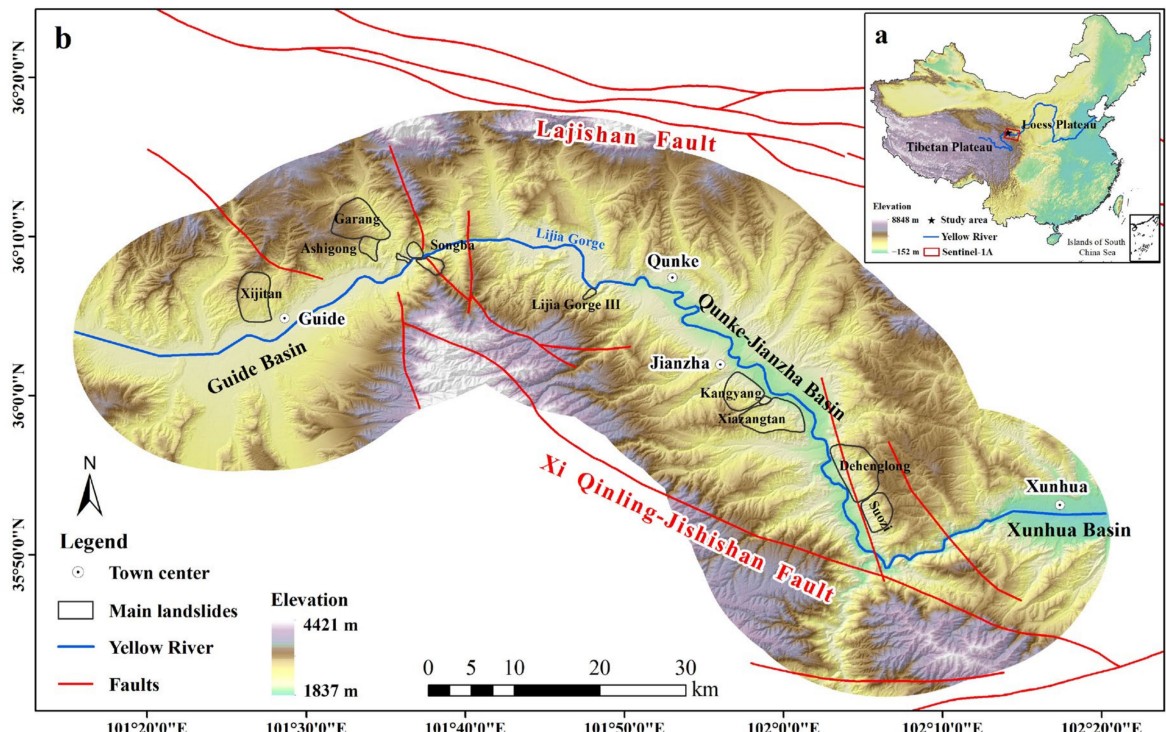

**Figure 1.** *Cont.*

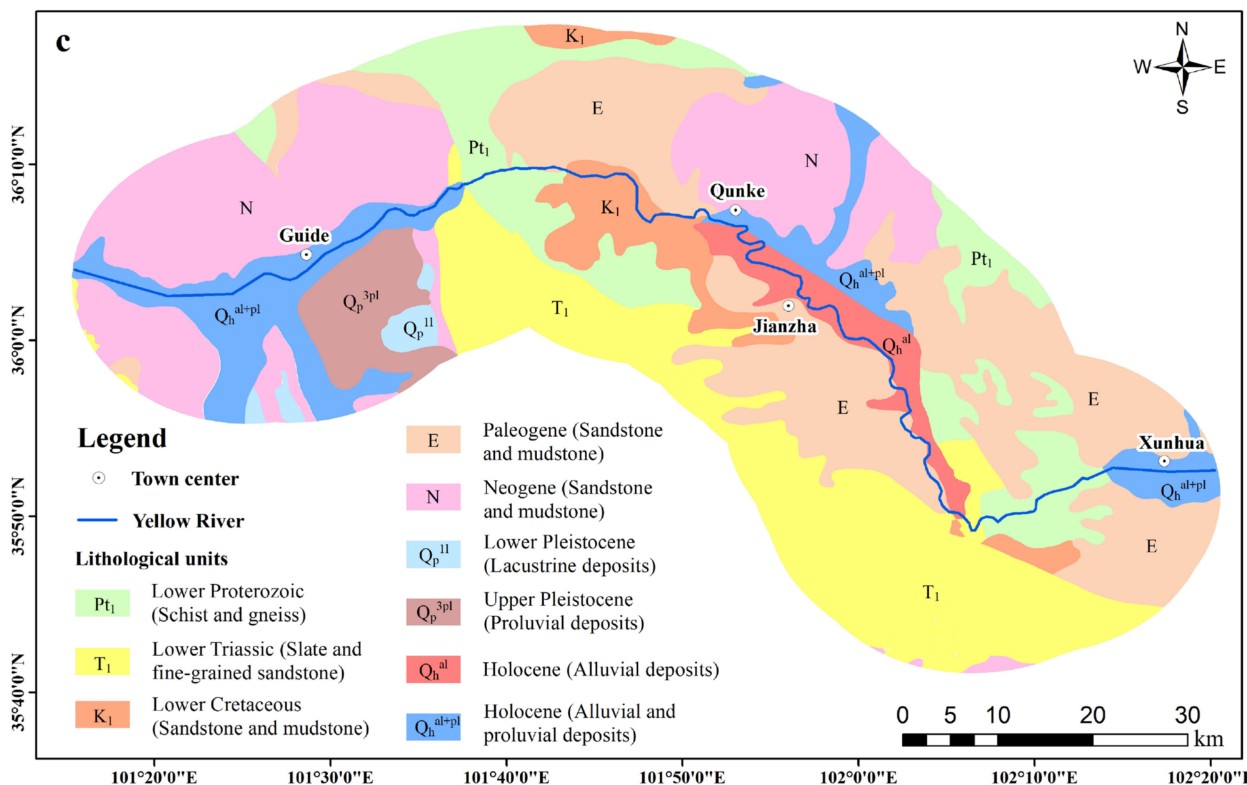

**Figure 1.** Location of the study area. (**a**) Geographical location of the upper reaches of the Yellow River; the red polygon indicates the area covered by Sentinel-1A; the black star indicates the location of the study area. (**b**) Topography environment, faults, and main landslides in the study area. (**c**) Geological map of the study area (collected from the China Geological Survey).

## 3. Datasets and Methodology

In this study, InSAR technology, hot spot analysis, and a semantic segmentation model were combined to extract the boundaries of the anomalous areas (Figure 2). Firstly, the displacement rate map of the study area was obtained using InSAR technology, and the anomalous areas were interpreted as labels manually based on the Google Earth image. Subsequently, a hot spot map obtained based on the hot spot analysis, Google Earth image, and topographic information were input into the deep learning semantic segmentation algorithms. In turn, the established semantic segmentation model enabled the identification of the boundaries of anomalous areas accurately.

### 3.1. Data

A Google Earth image captured on 11 March 2021 with a spatial resolution of 2.5 m (corresponding level 16 of Google Earth) was used to interpret the anomalous deformation areas and input as a feature image for training the model. In order to reduce the brightness, tones, and distortions difference of different Google Earth image tiles, the obtained Google Earth image was pre-processed, including color balance, haze removal, histogram equalization, a low pass filter, and a Gaussian filter based on ENVI (the Environment for Visualizing Images) software (v5.3). A total of 39 C-band (a wavelength of 5.63 cm) synthetic aperture radar (SAR) images of the Sentinel-1A satellite, spanning from 4 January 2020 to 28 April 2021, were collected to generate the displacement using InSAR technology. SAR images were captured along satellite track 135, in descending orbit, with an incidence angle of 39.2°. The imaging mode was the interference wide (IW) mode, and the polarization mode was vertical–vertical (VV) polarization. The spatial coverage of the SAR images is shown in Figure 1a. After the InSAR deformation monitoring, the coordinate system of the deformation vector was converted to be consistent with the Google Earth

image. The topography data including elevation, slope, and aspect, obtained from the Advanced Land Observing Satellite (ALOS) Phased Arrayed L-band Synthetic Aperture Radar (PALSAR) digital elevation model (DEM) with 12.5 m resolution, were used as feature images to analyze the semantic segmentation ability of the models. The cell size of the DEM was resampled to match the Google Earth image, and the coordinate system was converted to be consistent with the Google Earth image.

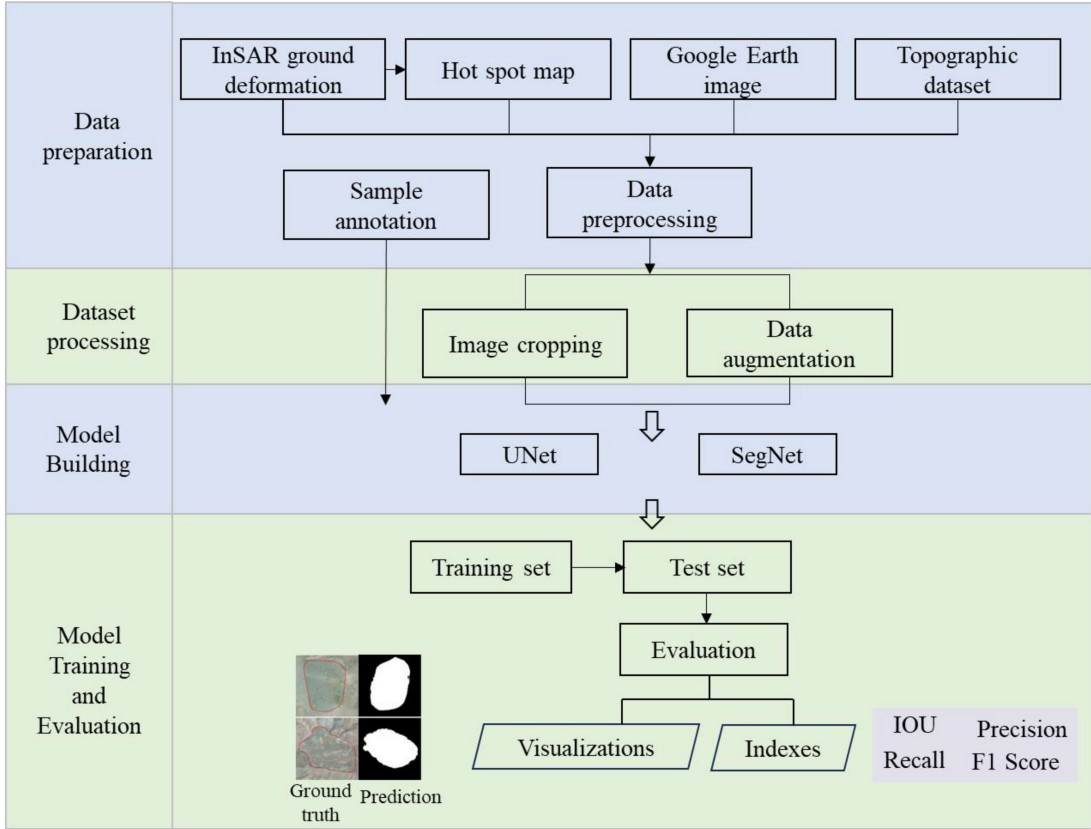

**Figure 2.** Workflow of the proposed method for identifying anomalous InSAR deformation areas. First, the input datasets, including the InSAR ground deformation map, hot spot map, Google Earth image, topographic dataset, and sample annotation are prepared and processed. Then, we selected the U-Net and SegNet models to train on the training set. Finally, a variety of metrics were adopted on the test set to evaluate the prediction performance of each model.

### 3.2. Interferometric Point Target Analysis (IPTA)

IPTA is a typical time series InSAR algorithm which can detect surface ground deformation with high precision (mm) for large areas and provide ground displacement rate maps associated with many geophysical processes [20,58]. IPTA, which has advantages in finding stable backscatters in areas of low coherence and can use large baselines for phase interpretation, was presented by [59]. The objective of IPTA is to identify target points with high quality which were determined through spectral diversity, amplitude dispersion, and standard deviation of the point target candidates [59].

First, all of the SAR images were co-registered to the selected reference scene (20200831), and then 71 interferograms were generated with a perpendicular baseline threshold of 160 m and a temporal baseline of 30 days (Figure 3). A reference point was selected to remove the residual terrain phase, which was determined by means of field investigation. After the differential interferograms were calculated, a two-dimensional regression analysis was performed to obtain height corrections, linear deformation rates, residual phases, and the unwrapped interferometric phase. Finally, the atmospheric phase and non-linear deformation phase were obtained from the residual phase by filtering the spatial

and temporal dimensions [59]. The coherence threshold was set as 0.7 to maintain a balance between the coverage density and ground deformation quality of the target points. An important aspect of the IPTA concept is the possibility of a step-wise, iterative improvement of different parameters. We processed the IPTA technology using GAMMA software (v20210701). Precise orbit data from the European Space Agency were used to correct the orbit error. The topographic phase can be removed by introducing external DEM. Two types of open-source DEM, obtained from 1-arc-second Shuttle Radar Topography Mission (SRTM) DEM with a resolution of 30 m and the ALOS PALSAR terrain corrected product with a resolution of 12.5 m, were used to compare the phase unwrapping results shown in Figure 4. Visually, both SRTM-1 and ALOS 12.5 have significant phase unwrapping errors, but ALOS 12.5 has extensive unwrapping null values in the lower right corner. The unwrapping percentages of SRTM-1 and ALOS 12.5 are 69.5% and 67.7%, respectively. The results indicate that SRTM-1 is more suitable for InSAR monitoring. As for why ALOS 12.5 has a worse effect, because 12.5 m is not the real resolution of the DEM, it is not generated from the PALSAR data itself, instead being obtained through interpolation after resampling from SRTM-1 [60,61]. It should not be used in place of a regular DEM because elevation values are altered by geoid corrections in preparation for radiometric terrain correction processing [61].

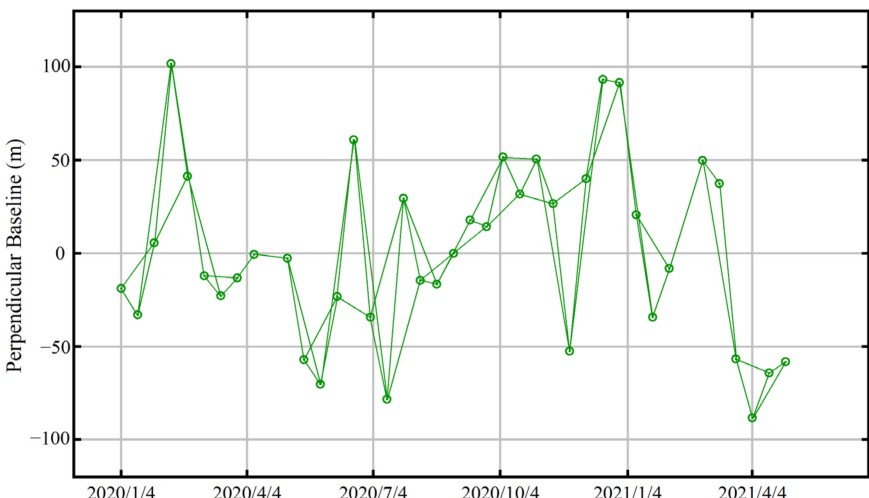

**Figure 3.** Spatial-temporal baselines of the generated interferograms. The numbered points represent SAR images and lines are the interferometric pairs used to form interferograms.

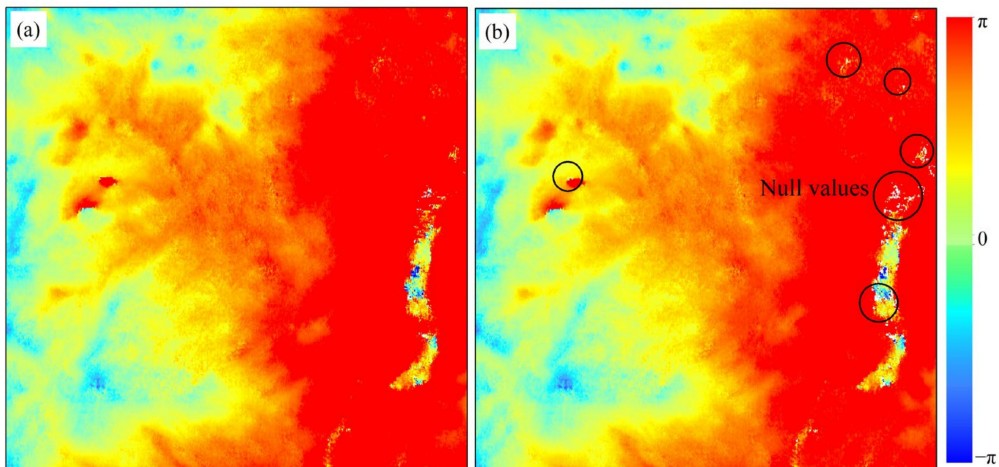

**Figure 4.** Phase unwrapping results based on two types of digital elevation models. (**a**) SRTM-1, and (**b**) ALOS 12.5.

*3.3. Hot Spot Analysis*

Hot spot analysis is a spatial statistics approach which can semi-automatically extract areas with large ground movement over mountain regions [62] and has been widely applied to identify and map anomalous areas with clustered high-velocity coherent targets (CTs) [30,63,64]. Hot spot analysis consists of the Getis–Ord Gi* statistic and kernel density estimation. The Getis–Ord Gi* statistic is a local spatial statistic model which was proposed to calculate the degree of aggregation of CTs within a specified distance, taking the deformation velocity as the weighting factor [65,66]. For each single CT at a site i and other CT at a site j, the Gi* can be calculated as follows:

$$\text{Gi}^* = \frac{\sum_{j=1}^{n} w_{i,j} * x_j - \overline{X} \sum_{j=1}^{n} w_{i,j}}{S * \sqrt{\frac{[n \sum_{j=1}^{n} w_{i,j}^2 - (\sum_{j=1}^{n} w_{i,j})^2]}{n-1}}} \tag{1}$$

where n is the total number of CTs; $w_{i,j}$ is the spatial weight with 1 for all CTs within a specified distance of CT i and 0 for other CTs including the CT i itself; $x_j$ represents the deformation velocity of each CT; $\overline{X}$ is the mean deformation velocity of all CTs; and S is the standard deviation of the deformation velocity of all CTs. In this study, the specified distance d was set as 150 m according to the scale of historical landslides in the study area and the spatial resolution of Sentinel 1A data. After the Getis–Ord Gi* statistic, the z-score (standard deviation) and *p*-value (independence probability) were calculated.

In order to highlight the distribution anomalous areas, the kernel density estimation was used, taking the z-score ass the weighting factor. Kernel density estimation can fit a smoothly tapered surface and can be used to calculate the density of the elements in a neighborhood around the elements [63,67]. The kernel density estimation can be calculated as follows [68]:

$$\text{Density} = \frac{1}{d^2} \sum_{i=1}^{n} \left[ \frac{3}{\pi} p_i \left( 1 - \left( \frac{x - x_i}{d} \right)^2 \right)^2 \right] \tag{2}$$

where d is the search radius; $x - x_i$ is the distance from each calculating pixel to CT i; n is the total number of CTs; and $p_i$ is the weighting factor for each CT. Finally, the anomalous areas were highlighted by several hot spots, which enable us to straightforwardly and easily understand the locations of anomalous areas. In this study, the hot spot analysis was implemented in the Spatial Statistics Tools in ArcGIS v10.6.

*3.4. Semantic Segmentation Models*

In this study, two popular deep learning models, namely U-Net and SegNet, were used to compare their performance in extracting the boundaries of anomalous areas.

3.4.1. U-Net Model

The U-Net model [69] is widely applied to semantic segmentation tasks owing to its robust network structure, fast training speed, and suitability for small data sets [70,71]. The U-Net model is a deep learning framework based on fully convolutional networks and has an encoder–decoder architecture similar to the letter "U". The U-Net structure used, as illustrated in Figure 5, comprises a contracting path (an encoder) for extracting main and complex features and an expanding path (a decoder) for retrieving the relevant spatial location [72]. In the encoder, the convolution and max pooling layers are combined, which follows the typical CNN architecture to extract the characteristics of feature images including the spectral information and texture in the spatial dimension [72]. The encoder path is composed of several blocks of two convolution layers with a 3 × 3 kernel size, each followed by a Rectified Linear Unit (ReLU) activation function and a max-pooling layer with a size of 2 × 2. After a max-pooling operation, the spatial dimension of the feature maps is reduced to half of the input size. In the decoder, the transposed convolution and skip connection methods were applied to retain the dimensions and information that were lost during down-sampling [70]. The decoder path is composed of several blocks with a

transpose convolutional layer of size 2 × 2 and two convolutional layers with a kernel size of 3 × 3 each followed by ReLU as an activation function. Finally, a convolutional layer with a 1 × 1 kernel size with a sigmoid function is added to output the segmentation result. The most important part of the U-Net architecture is the skip connections between the encoder and decoder stages. The output from each block before max pooling in the encoder part is concatenated with the symmetrical position in the decode part, which can retrieve the relevant spatial information lost in the encoder path [73].

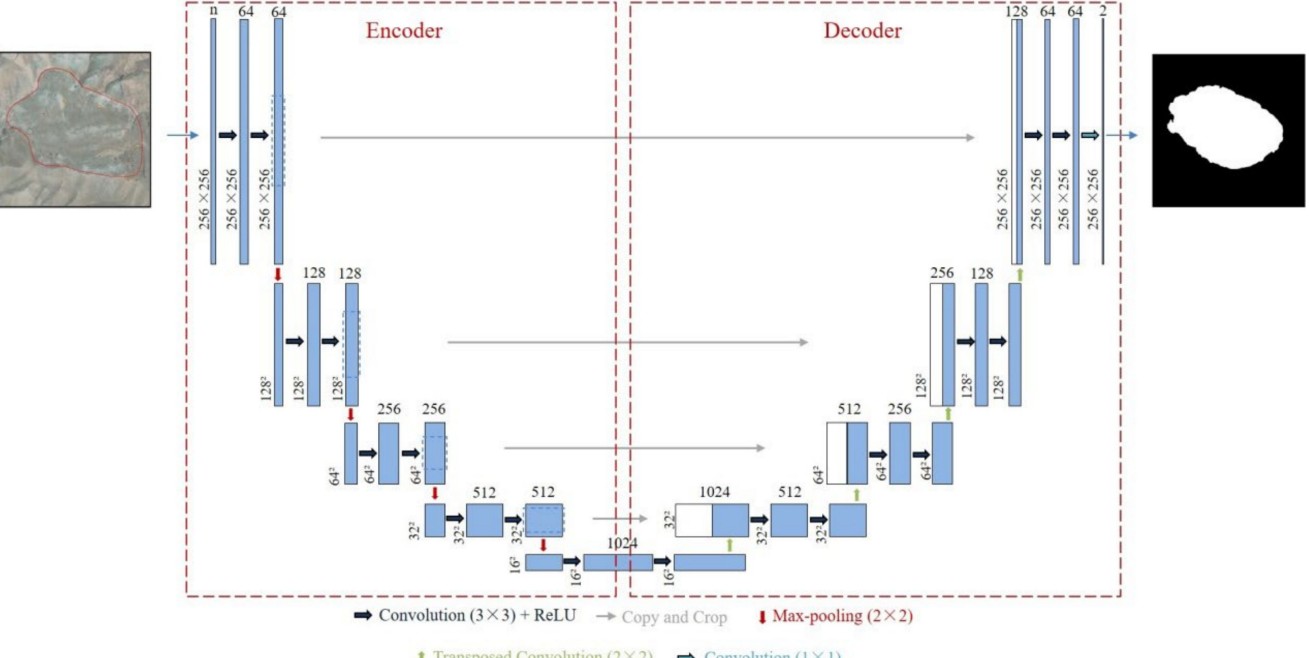

**Figure 5.** Illustration of the U-Net architecture. The number of channels is indicated at the top of the box, where n represents the number of input image channels. The x–y size is shown in the lower left corner of the box. White boxes represent maps of copied resources. Modified from [69].

### 3.4.2. SegNet Model

The SegNet model is another well-known encoder–decoder network for image semantic segmentation [36,74]. The architecture of SegNet is illustrated in Figure 6, which consists of an encoder portion and a corresponding decoder portion [75]. The encoder portion is used to train the segmentation engine, while the decoder portion is used to obtain pixel-wise classification [74,76]. Each encoder contains 3 × 3 convolution layer followed by a batch normalization layer and a ReLU activation function to produce a set of feature maps. Each encoder repeats the implementation of this combination of convolution, batch normalization, and ReLU activation multiple times in each encoder layer. At the end of each encoder layer, a 2 × 2 max-pooling layer with stride of 2 is applied to downsample the output feature map by 2, and the number of feature channels is doubled at each down-sampling step. A total of 13 convolution layers are added in the encoder part, which resembles the VGG16 network [77]. At each decoder layer, the number of feature channels is reduced by half, while the size of the output feature map is doubled by using a 2 × 2 inverse max-pooling layer with stride of 2 where the memorized max-pooling indices are received to up-sample the feature maps to a larger scale [74,78]. After performing up-sampling, these feature maps are applied with repeated multiple 3 × 3 convolution layers followed by a batch normalization layer and a ReLU activation function. Finally, the high dimensional feature map at the output of the final decoder is applied with the softmax function to classify the image into two classes.

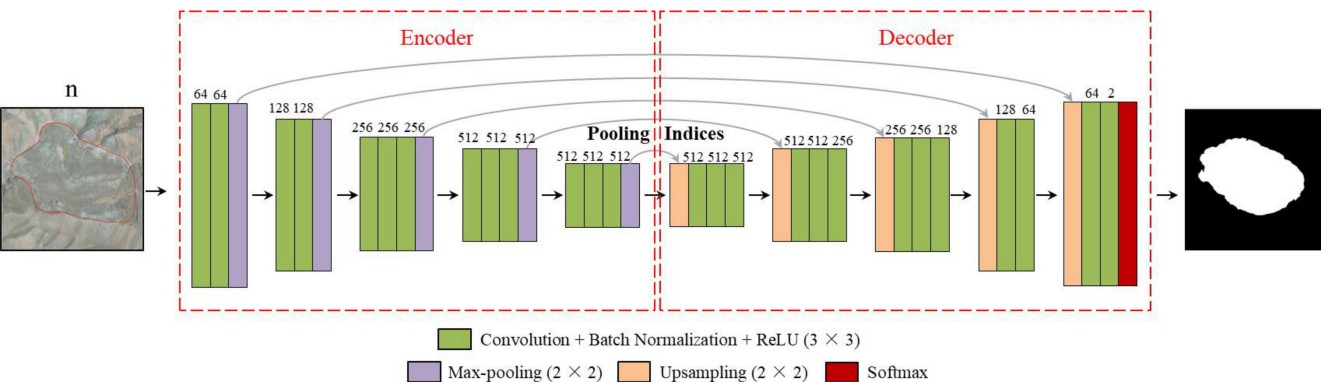

**Figure 6.** Illustration of the SegNet architecture. The number of channels is indicated at the top of the box, where n represents the number of input image channels. Modified from [74].

*3.5. Evaluation Indexes*

To evaluate the performance of the built models, evaluation indexes including recall, precision, F1-score, and Intersection over Union (IoU) were used as quantitative metrics to evaluate the extraction results. Precision represents the ratio of pixels of correctly predicted anomalous areas to the total number of pixels of predicted anomalous areas, and recall is the ratio of pixels of correctly predicted anomalous areas to the labels annotated. The F1-score is the harmonic average between precision and recall, which attempts to takes into account both precision and recall [36,79]. The IoU computes the overlapping of areas between the label annotation and the model extraction. These evaluation indexes are defined as follows [36]:

$$\text{Precision} = \frac{T_P}{T_P + F_P} \tag{3}$$

$$\text{Recall} = \frac{T_P}{T_P + F_n} \tag{4}$$

$$F_1 = \frac{2 \times \text{Precision} \times \text{Recall}}{\text{Precision} + \text{Recall}} \tag{5}$$

$$\text{IoU} = \frac{A \cap B}{A \cup B} = \frac{TP}{TP + FP + FN} \tag{6}$$

where $T_P$ is true positive, which represents the pixels that were correctly detected as anomalous areas; $F_P$ is false positive, which represents the pixels that were incorrectly detected as anomalous areas; and $F_N$ is false negative, which represents the pixels that were incorrectly detected as non-anomalous areas.

## 4. Experiment and Results

*4.1. Dataset Preparation*

4.1.1. InSAR Deformation

The mean displacement rate map along the direction of radar line-of-sight (LOS) obtained from the descending Sentinel datasets is illustrated in Figure 7. We obtained 930,193 CTs covering the study area with an average density of 264 CTs/km², which is sufficient to detect potential landslides using InSAR technology. However, in the southern part of Lijiaxia reservoir, an absence of CTs was observed in the high-altitude mountainous areas, resulting in geometric distortion and low coherence of the radar signal. The LOS displacement rate ranged from −234 to 366 mm/y, the negative values representing ground motion away from the satellite, and the positive values representing movement towards the satellite. In practice, the stability threshold of InSAR displacement rates is determined based on the standard deviation of the velocity of the whole region [80,81], so a stability

threshold of ±10 mm/y was chosen to identify anomalous areas which were regarded as the possible locations of potential landslides.

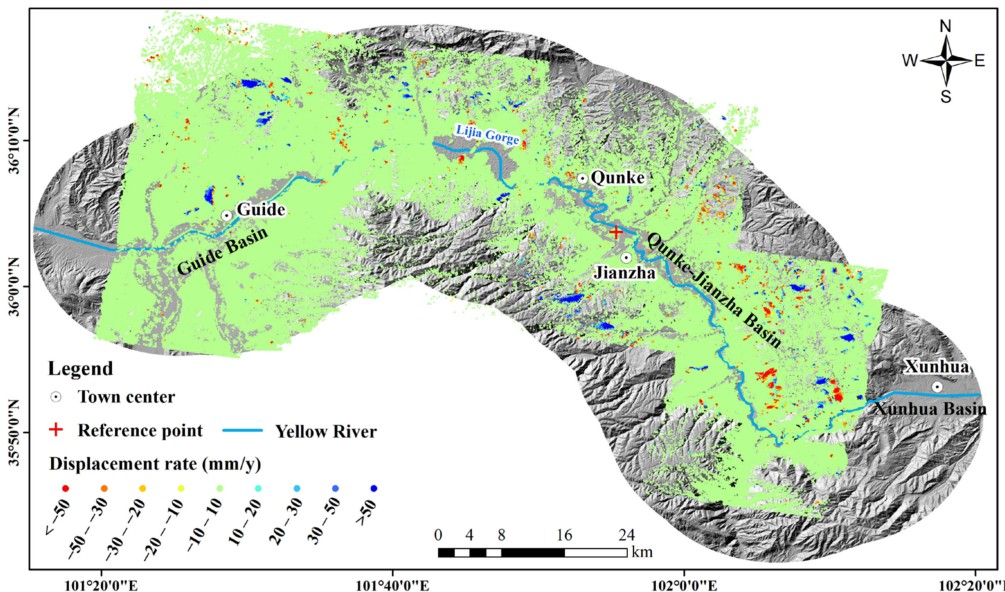

**Figure 7.** LOS displacement rate derived by IPTA technique. A positive value indicates a point moving toward the satellite and a negative value indicates a point moving away from the satellite.

In the experiment, the InSAR deformation result was used as the input feature by converting the CTs into a raster dataset with a value of −234 to 366 in ArcGIS (Figure 8). The locations of the CTs were replaced by cells and the values of the cells were determined based on the displacement rate of the CTs. The value of cells without CTs located in them was set as null value. In order to match the cells in the output raster alignment of the Google Earth image raster, the output cell size was set to be consistent with the Google Earth image and the function of the snap raster was set. Finally, the InSAR deformation raster was used as a band combined with the Google Earth image to build the semantic segmentation model.

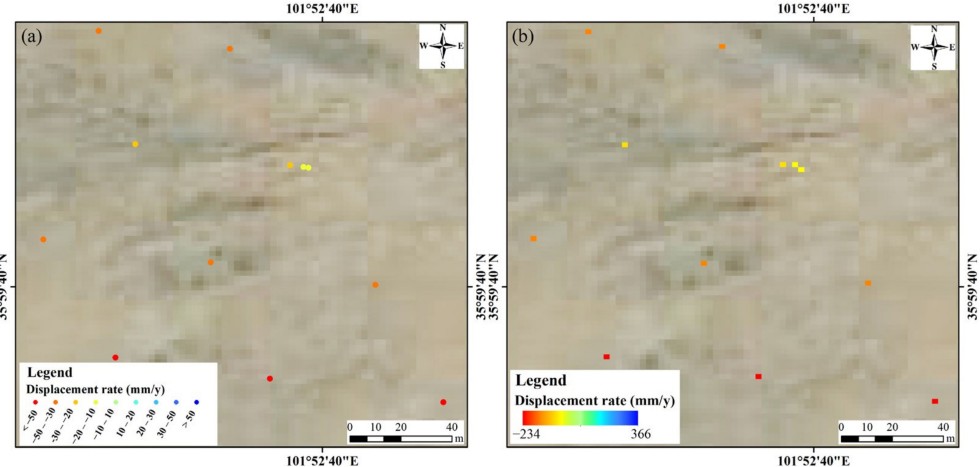

**Figure 8.** Example of conversion results from coherent targets into a raster dataset. (**a**) Coherent targets distribution, and (**b**) raster dataset distribution.

4.1.2. Hot Spot Analysis

The output of hot spot analysis is a hot spot map, as shown in Figure 9, which can highlight areas with anomalous deformation [29]. The red spots indicate the areas with

clustered high velocity CTs moving towards the sensor, whereas blue spots indicate the areas with clustered high velocity CTs moving away from the sensor. The green color indicates that there are no clustered CTs in these areas. The detected anomalous areas were characterized by intense ground deformation and a significant aggregation degree, which is similar to the process of manually extracting of anomalous areas based on InSAR deformation. Compared to a traditional InSAR map, the hot spot map can rapidly extract useful information from a large amount of CTs.

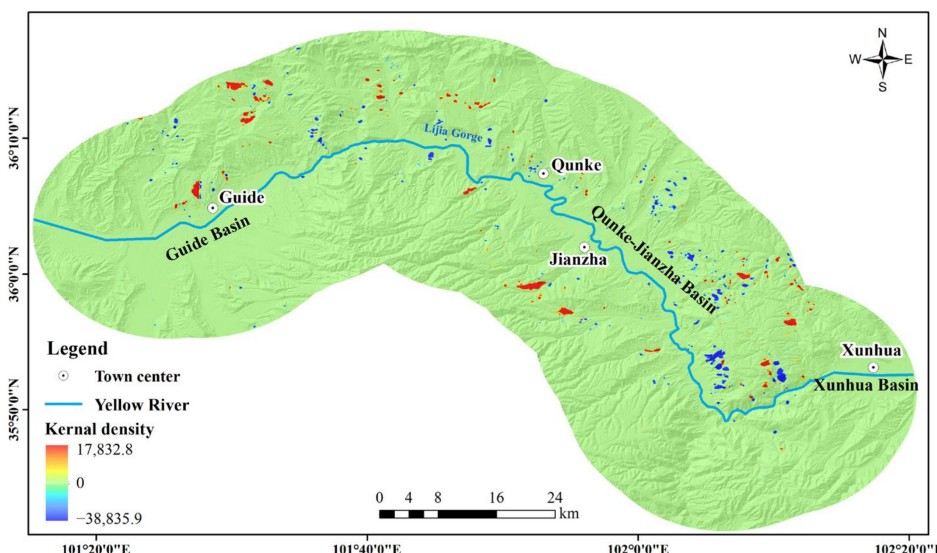

**Figure 9.** The hot spot map derived from hot spot analysis using the InSAR displacement rate map. The red spots correspond to the anomalous areas with high velocity moving towards the sensor, whereas blue spots correspond to the anomalous areas with high velocity moving away from the sensor.

In the experiment, the hot spot map with a value of −38,835.9 to 17,832.8 was used as an input feature combined with the Google Earth image to build the semantic segmentation model. In order to match the cells of the hot spot map with the raster alignment of the Google Earth image raster, the output cell size in the kernel density estimation was set to be consistent with the Google Earth image and the function of the snap raster was set.

### 4.1.3. Topographic Dataset

The elevation, slope, and aspect derived from DEM were input as feature images into the semantic segmentation models. The channels of elevation, slope, and aspect have values of 1837 to 4421, 0 to 90, and 0 to 360, respectively. The topographic dataset contains more spatial information which can improve the extraction accuracy of the anomalous deformation areas [41,42]. In the experiment, the topographic dataset was combined with the optical image and deformation dataset as feature images to build the semantic segmentation model.

### 4.1.4. Sample Annotation

The anomalous deformation areas were manually interpreted and outlined based on the InSAR displacement rate map and the Google Earth image using the ArcGIS software. The development of landslides had the following image characteristics. (1) There are abnormal arc shapes developed on the rear margin of the landslide body, including "round chair shaped" and "dustpan-shaped" landslide back wall steep ridges [82], which show differences in texture, color, brightness (Figure 10a,b). (2) Fragmented slope surface, poor vegetation cover, and obvious cracks and scarps indicate the movement signs (Figure 10a,b) [83,84]. (3) The landslide front edges often show flat terrain of landslide deposits and were distinguished by differences in slope and aspect (Figure 10c). The

boundaries of the deformation areas were interpreted initially via the proposed image characteristics. Then, the boundaries of anomalous deformation areas were verified and modified accurately based on the field survey results. To ensure sample annotation quality, three landslide experts conducted cross-validation of the anomalous deformation area interpretation results. In total, 476 anomalous deformation areas were obtained. Finally, the label dataset was obtained by converting the deformation area inventory maps into a raster dataset with the same resolution as the Google Earth image. The binary map is shown in Figure 11, where the anomalous areas are encoded as one and the non-anomalous areas are encoded as zero.

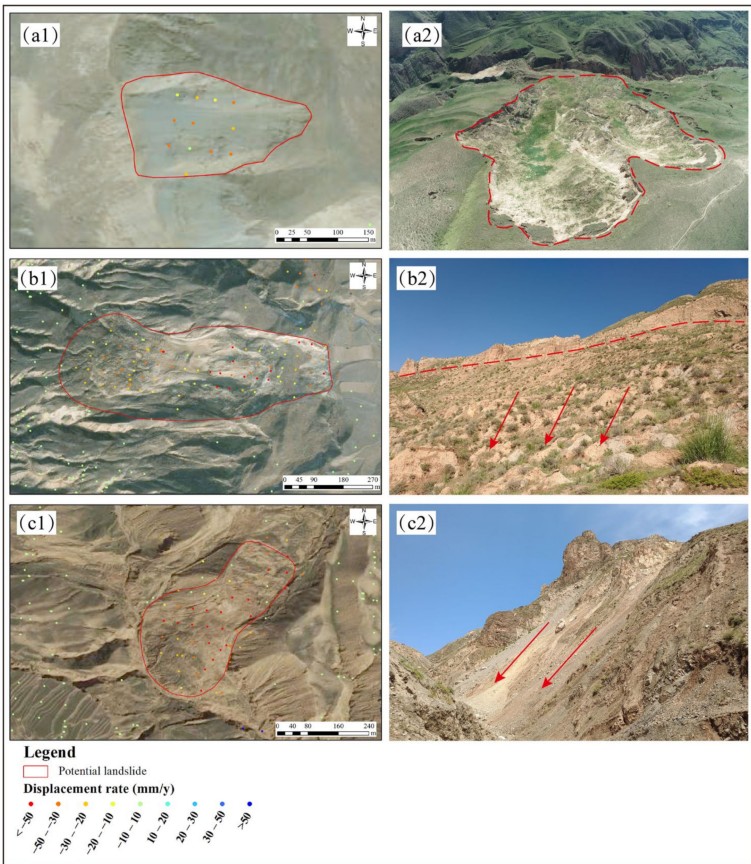

**Figure 10.** Mean displacement of potential landslides and the boundaries of the anomalous deformation areas were determined by means of field survey. (**a1**,**b1**,**c1**) Boundaries of anomalous deformation areas determined from InSAR deformation map and Google Earth image. (**a2**,**b2**,**c2**) Photographs of anomalous deformation areas in the field investigation. Red solid lines in (**a1**,**b1**,**c1**) indicate the boundaries of anomalous deformation areas. Red arrows indicate the direction of slope movement. Red dashed line in (**a2**) indicates the boundary of anomalous deformation areas in (**a1**). Red dashed line in (**b2**) indicates the location of scarp.

### 4.2. Dataset Processing for Model

Considering the limitation of computer memory conditions, the image size of the feature images is 50,661 × 35,415 pixels, making it impossible to input these large-size images for training directly. We thus scanned the image using a sliding window algorithm to generate training patches, as shown in Figure 12. As the size of the anomalous deformation areas was concentrated in the 48 × 56–256 × 256 pixel range, the size of the cropped images for model was set as 256 × 256 pixels. This process can speed up the training and has little effect on the results of extracting the boundaries of anomalous areas [43]. In this study, the same method was used to split the feature and label images with a 20% overlap, and a total of 42,904 feature images were obtained.

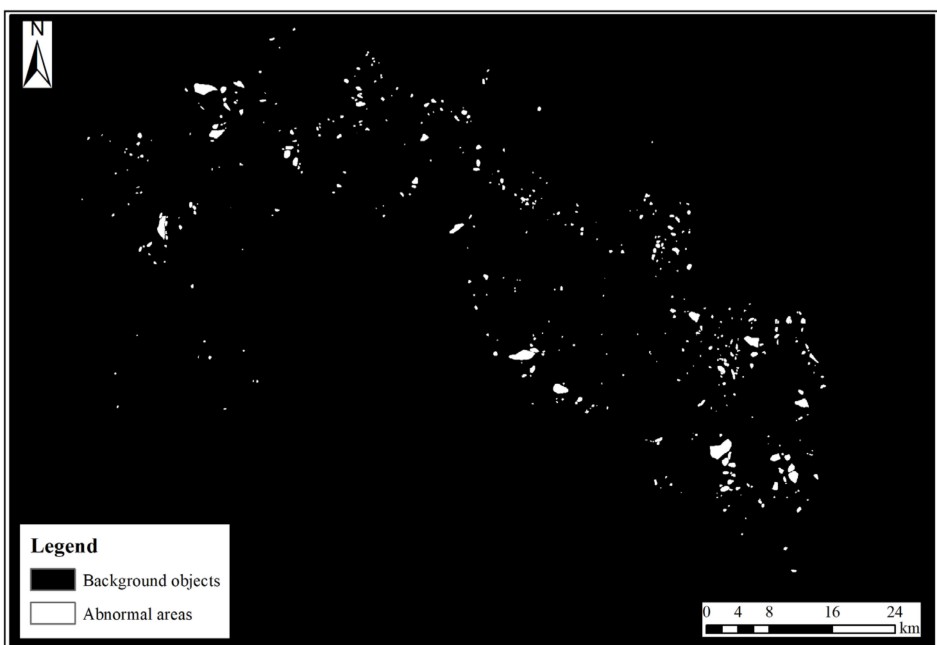

**Figure 11.** The binary map of anomalous deformation areas.

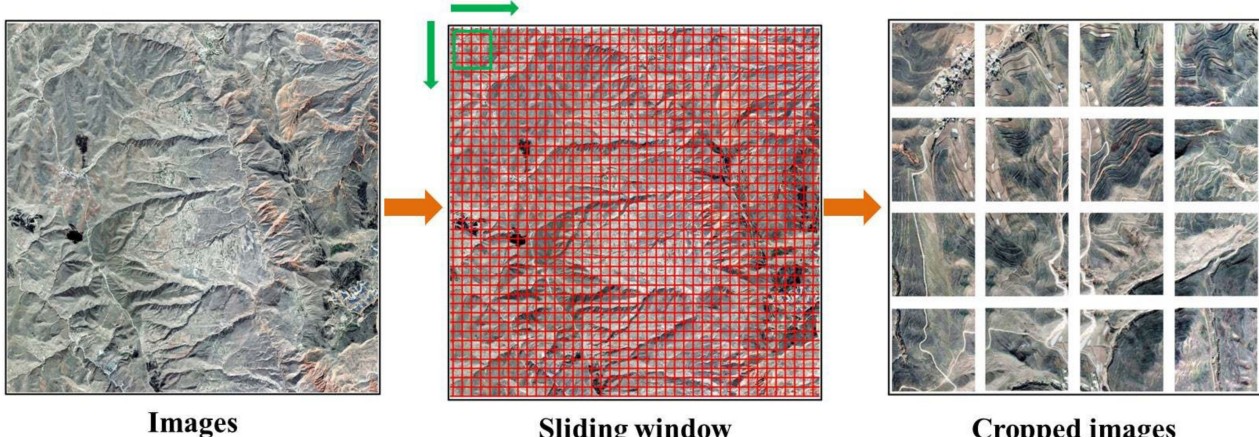

**Figure 12.** Flowchart of cropping feature images through scanning the image using a sliding window. Orange arrows indicate the steps of cropping feature images. Green arrows indicate the scanning directions of sliding window.

Different input features often have different dimensional units, which will affect the results of model training. To eliminate the influence between features, image normalization is performed to make the input in the same order of magnitude. Each channel of the feature images was normalized to 0–1 via Min–Max normalization [85]. In particular, the accuracy of deep learning models can be determined by the quantity and diversity of data, which can be significantly increased using data augmentation methods. In this study, data augmentation is employed for the semantic segmentation models to learn more intrinsic and invariant characteristics related to the boundaries of anomalous areas. Data augmentation methods including random horizontal and vertical flips and random rotations were applied randomly to the training dataset (Figure 13).

In this study, we used a small-batch training strategy, which only requires a small portion of the training dataset in each iteration to avoid local minimization of training errors and achieve rapid convergence in the parameter optimization process [86]. Finally, in total, 70% of the feature images (30,032) were randomly selected as training dataset to train

the models. The remaining 30% of the feature images (12,872) were taken as the testing dataset to evaluate the performance of the models, and all the networks shared the same dataset processing for training and testing.

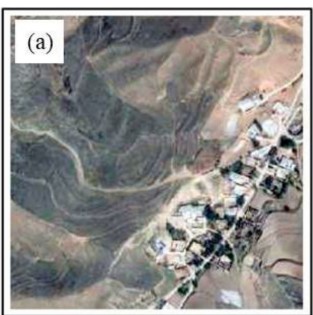 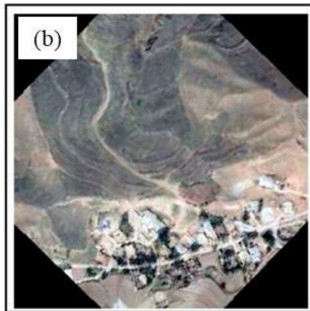 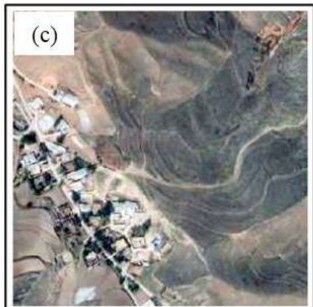 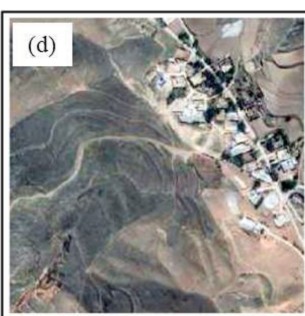

**Figure 13.** Schematic diagram of the data augmentation. (**a**) Original image, (**b**) random rotation of 90°, (**c**) horizontal flip, and (**d**) vertical flip.

*4.3. Experimental Setup*

The experiments were implemented in a computer with the configuration of AMD Ryzen 9 5900X (3.70 GHz) (made in Suzhou, China) and NVIDIA GeForce RTX 3090 (made in Guangzhou, China), and Pytorch (GPU version) was adopted as the deep learning framework. The Adam optimizer was used to update and optimize the parameters of models. The number of training epochs was set as 30 with a batch size of 8, and the iterative training on the training set can continuously reduce the loss function value to optimize model parameters [87]. The initial learning rate of training was set to $1 \times 10^{-3}$. Binary Cross-Entropy Loss (BCE Loss) was chosen as the loss function for training, which is defined as follows:

$$\text{loss} = -\frac{1}{N} \sum_{i=1}^{N} y_i \times \log(p(y_i)) + (1 - y_i) \times \log(1 - p(y_i)) \tag{7}$$

where N denotes the number of classification categories, and p represents the probability that the label category of $x_i$ is $y_i = 1$. A threshold of 0.5 was adopted to classify the probability of the grids belonging to the boundaries of the anomalous areas.

*4.4. Experimental Result*

4.4.1. Comparison of Different Deformation Indexes

Firstly, the performance of different deformation indexes in two models was evaluated quantitatively according to the evaluation indexes as mentioned above. The precision, recall, F1 score, and IoU of the models are shown in Table 1. Model I performs worse than other models, especially in the terms of recall, F1 score, and IoU, which indicates poor segmentation of anomalous areas. Comparing model II's performance based on the Google Earth image combined with the hot spot map as input features, the precision, recall, F1 score, and IoU are 0.853, 0.779, 0.809, and 0.686, respectively, which indicates that the hot spot map has a better guiding effect to extract anomalous areas than InSAR deformation points. The F1 score of model IV reaches 0.812, which is higher than those of the other three models. Although the precision of model IV is not the highest, the recall and IoU of model IV are significantly higher, which can reflect the extent of capture of the target pixel [88]. The performance of the U-Net model is better than that of the SegNet model, regardless of the two different deformation indexes, which means that the U-Net model is more suitable for obtaining abstract information. The difference in the performance of U-Net and SegNet model may be attributed to the skip connection of U-Net that incorporates the low-level and high-level features so that the decoding path has more available spatial information to improve the segmentation performance [36]. Although InSAR deformation points can be widely applied to delineate high deformation areas manually by scholars, which is

consistent with the human understanding of deformation areas, the hot spot map can specifically reflect the boundaries and locations of anomalous deformation areas and can more easily be understood by algorithms.

**Table 1.** The anomalous areas detection results of different models and different input features in testing dataset.

| Num | Model | Data | Precision | Recall | F1 Score | IoU |
|---|---|---|---|---|---|---|
| I | SegNet | Google Earth Image (RGB) + InSAR Deformation Points | 0.879 | 0.719 | 0.785 | 0.652 |
| II | | Google Earth Image (RGB) + Hot Spot Map | 0.853 | 0.779 | 0.809 | 0.686 |
| III | U-Net | Google Earth Image (RGB) + InSAR Deformation Points | 0.843 | 0.775 | 0.806 | 0.676 |
| IV | | Google Earth Image (RGB) + Hot Spot Map | **0.829** | **0.807** | **0.812** | **0.689** |

### 4.4.2. Topographic Information Addition Identification Model

The topographic information (elevation, slope, and aspect) can be supplementary to some confusing textures and shapes in optical images, which could help to provide a good distinction of slope boundaries [89], and as such has been widely applied in landslide semantic segmentation [36,43,89]. The quantitative evaluation results of introducing different combinations of topographic information into the detection models are shown in Table 2. It is obvious that adding slope and aspect can significantly improve the segmentation performance of models. However, elevation may contribute to new confusion, potentially even leading to the performance of models decreasing compared to before adding elevation. This effect is also present in other combinations of topographic information which involve elevation. The optimal combination of topographic information is to only add aspect with the Google Earth image and hot spot map as input features. Compared with model IV, model XIII can obviously improve the performance of anomalous area detection, with an increase in recall, F1 score, and IoU of 2.8%, 1%, and 1.6%, respectively.

**Table 2.** The anomalous areas identification results of different combinations of topographic information addition identification models in testing dataset.

| Num | Model | Data | Precision | Recall | F1 Score | IoU |
|---|---|---|---|---|---|---|
| V | | Google Earth Image (RGB) + Hot Spot Map + Slope | 0.797 | 0.836 | 0.811 | 0.688 |
| VI | | Google Earth Image (RGB) + Hot Spot Map + Aspect | **0.816** | **0.833** | **0.819** | **0.7** |
| VII | | Google Earth Image (RGB) + Hot Spot Map + Elevation | 0.847 | 0.758 | 0.794 | 0.666 |
| VIII | SegNet | Google Earth Image (RGB) + Hot Spot Map + Slope + Aspect | 0.806 | 0.839 | 0.817 | 0.696 |
| IX | | Google Earth Image (RGB) + Hot Spot Map + Slope + Elevation | 0.82 | 0.814 | 0.811 | 0.689 |

**Table 2.** *Cont.*

| Num | Model | Data | Precision | Recall | F1 Score | IoU |
|---|---|---|---|---|---|---|
| X | | Google Earth Image (RGB) + Hot Spot Map + Aspect + Elevation | 0.828 | 0.797 | 0.806 | 0.682 |
| XI | | Google Earth Image (RGB) + Hot Spot Map + Slope + Aspect + Elevation | 0.846 | 0.78 | 0.806 | 0.682 |
| XII | | Google Earth Image (RGB) + Hot Spot Map + Slope | 0.828 | 0.812 | 0.814 | 0.693 |
| XIII | | Google Earth Image (RGB) + Hot Spot Map + Aspect | **0.822** | **0.835** | **0.823** | **0.705** |
| XIV | | Google Earth Image (RGB) + Hot Spot Map + Elevation | 0.773 | 0.823 | 0.792 | 0.662 |
| XV | U-Net | Google Earth Image (RGB) + Hot Spot Map + Slope + Aspect | 0.81 | 0.836 | 0.818 | 0.697 |
| XVI | | Google Earth Image (RGB) + Hot Spot Map + Slope + Elevation | 0.858 | 0.765 | 0.803 | 0.678 |
| XVII | | Google Earth Image (RGB) + Hot Spot Map + Aspect + Elevation | 0.849 | 0.786 | 0.811 | 0.688 |
| XVIII | | Google Earth Image (RGB) + Hot Spot Map + Slope + Aspect + Elevation | 0.811 | 0.836 | 0.818 | 0.698 |

The U-Net and SegNet model have different predictive abilities, in that the former has a fast convergence speed and can achieve good predictive results based on a small sample set [69,90]. SegNet can obtain good predictive results when the number of samples is huge, and SegNet deeply depends on the accuracy of the labels [72,87]. However, the anomalous deformation areas were manually interpreted and outlined based on the InSAR displacement rate map and Google Earth image, which means that the dataset annotation result may be slightly different between different researchers, meaning that the U-Net model is more suitable for extracting the boundaries of anomalous areas.

Adding topographic information can provide the spatial characteristics of slopes, which is helpful to identify the boundaries of anomalous deformation areas based on the topographic characteristics [36,91]. Aspect has good consistency in small areas of topography and can be considered as an effective feature to identify anomalous deformation areas [92]. Figure 14 shows some identification results of different models based on the test set. It is obvious that topographic information can constrain the boundaries of anomalous deformation areas and is more similar to the actual labels (Figure 14c). Although the identified results of different models are still incomplete, adding topographic information can expand the identified areas and leads to the process being more like manual interpretation.

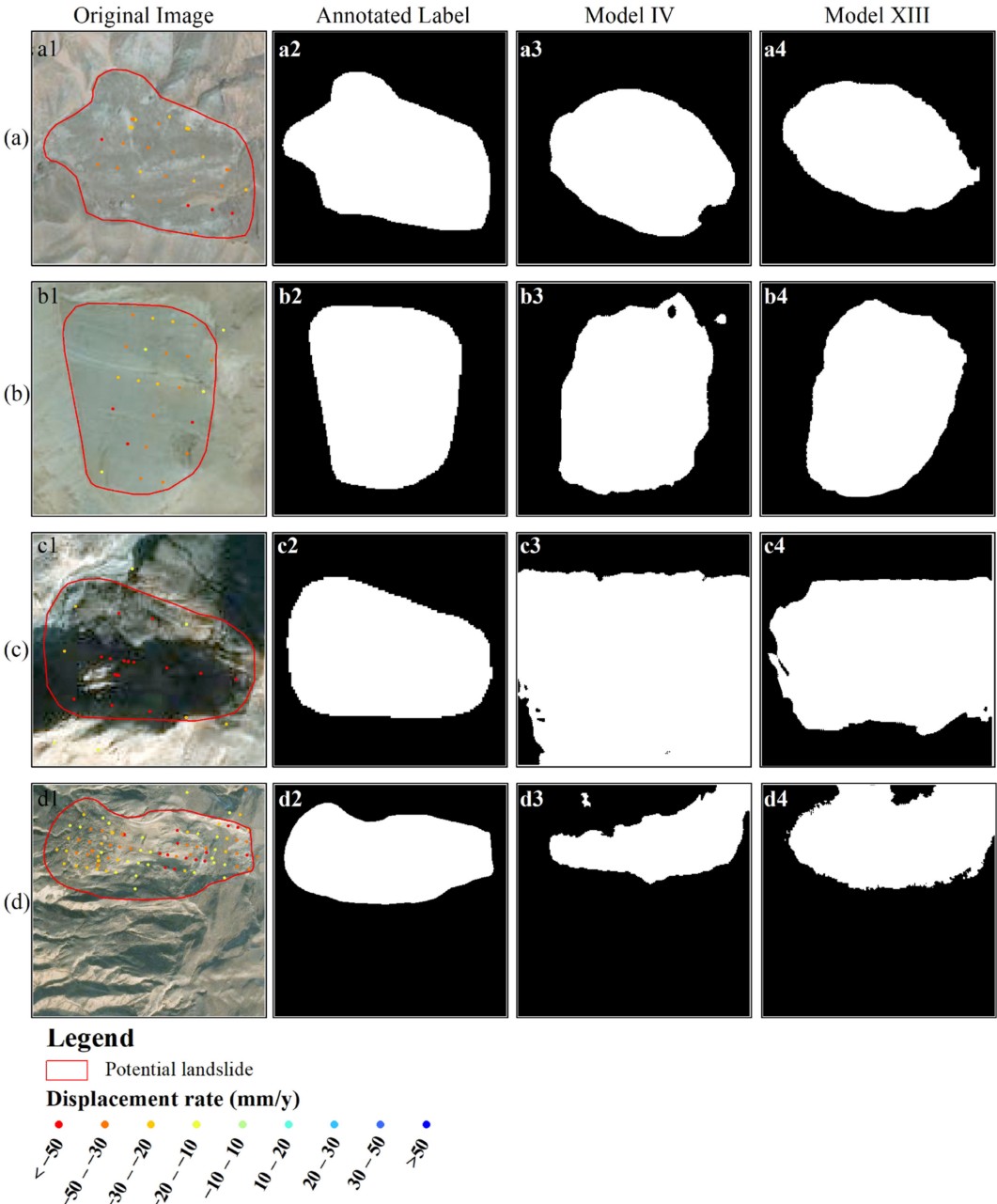

**Figure 14.** Identification results of different models on the testing dataset (**a–d**). The four images in each row represent the original image, annotated label, identification result of model IV, and identification result of model XIII, respectively. Red lines in (**a1,b1,c1,d1**) indicate the boundaries of anomalous deformation areas.

## 5. Discussion

### 5.1. Effects of the Data for Model

In this study, the data used to establish semantic segmentation models includes different deformation indexes and different combinations of topographic information. In terms of the deformation indexes, the hot spot map is more suitable for the semantic segmentation model in identifying anomalous deformation areas. The hot spot map can highlight the location of anomalous deformation areas, which is similar to delineating anomalous deformation areas manually. In terms of the different combinations of topographic information, aspect and slope may play an important role in distinguishing slope boundaries, but elevation may lead to new confusion. In a previous work, slope, aspect, and elevation were

combined as topographic information input into a semantic segmentation model [43,89], which is unnecessary and leads to complex data processing and training. The basis of delineating the boundaries of anomalous deformation areas is similar to slope unit division, which involves maximizing the homogeneity within each extracting boundary and maximizing the heterogeneity between inside and outside the anomalous area's deformation boundaries [93,94]. The terrain aspect, defined as the projected direction of a line normal to the slope on the horizontal plane, can be used to identify the direction of the steepest descent at a location on the ground surface [92]. The terrain aspect for the actual natural slopes can be considered similar and homogeneous. Inputting slope may lead to the neglect of some micro-topographic features and a more complex feature analysis process [92]. It is advised to introduce aspect as topographic information to extract anomalous deformation areas, though more experimental results from high-resolution DEM are needed to support this opinion.

### 5.2. Advantages of the Model

Delineating anomalous deformation areas derived from InSAR results is an important part of mapping potential active landslides over a large scale; the dominant method in this process is still manual interpretation through human–computer interaction, which is time- and labor-consuming. In this study, the semantic segmentation deep learning model combined with Google Earth images, InSAR deformation hot spot map, and topographic information was used to identify the boundaries of anomalous deformation areas automatically. Compared with existing research about delineating InSAR anomalous deformation areas, the proposed method can more accurately extract the anomalous deformation areas, owning to its special input features and advanced identification methodology. In terms of the input features, InSAR deformation results can be input into the prediction model in different forms. The widely used InSAR deformation forms include time series InSAR deformation and differential InSAR deformation [95]. However, directly using InSAR deformation may ignore the effect of scattered high-deformation areas induced by data-processing errors, which are excluded during visual interpretation. The hot spot map has similar characteristics to delineating anomalous areas manually, which only identifies clustering high ground deformation and is helpful for decreasing the effect of data-processing errors. The hot spot map also contains the information of the deformation stability threshold, avoiding the influence of widely distributed stable areas. Moreover, adding topographic information is similar to providing three-dimensional images for research to delineate anomalous deformation areas. The identification result would be more inclined to provide a good distinction of slope boundaries, which is consistent with other studies [43,89].

In terms of identification methodology, OBIA technology was used to extract anomalous areas automatically [31], but the implementation process needs professional practical ability to determine the optimal segmentation scale, select suitable features, and establish the classification ruleset [96]. This method is a semiautomated interpretation approach for anomalous deformation areas identification [31]. Too many factors would affect the extraction accuracy in the process of OBIA, leading to the shortcoming of low robustness and generalization ability [97]. Existing models are often unable to obtain good extraction results for different study areas [35]. Unlike the OBIA method, the semantic segmentation deep learning method is an end-to-end algorithm that can directly input image information as a supervised signal to classify segmented pixels [98]. It can automatically extract the most relevant features of the target task depending on loss function, which has the advantages of solid robustness [31,99]. There is no need to select segmentation parameters and combine multi-dimensional features manually. Although the existing deep learning model is complex, with the internal decision-making method being opaque, poorly interpretable, or un-interpretable, appearing like a "black box", the end-to-end learning approach has the advantage of making it easier to obtain the global optimal solution [35]. Based on the semantic segmentation deep learning method, an efficient and high-precision classification

of remote sensing images can be realized [35]. This method can provide effective, real-time, and high-precision classification results for potential landslide identification.

*5.3. Limitations and Further Directions of the Model*

There are some limitations regarding input features and model establishment that need to be considered in future research. Firstly, an accurate ground deformation result is the most basic step in the research. The areas with insufficient anomalous deformation points, induced by spatial and temporal decorrelation [100], would be excluded by the method, leading to incompletely delineating deformation areas. This may be dealt with by using long-wavelength SAR acquisitions at high spatial and temporal resolutions, which have better penetrability and enable the effective monitoring of ground deformation in complex mountainous areas [101]. In addition, the spatial resolution of the topographic dataset can affect the details of geomorphological characteristics. Depending on the scale of the landslides in the study areas, the used spatial resolution of the topographic dataset can be determined. More precise extraction of small-scale slopes requires combining higher-resolution topographic information based on unmanned aerial vehicles (UAV) or other high-precision sensors [43]. Secondly, with the development of advanced semantic segmentation deep learning models, the extraction result of anomalous deformation areas will be improved, such as adding an attention module [102] or combining other popular structures [103]. Moreover, using fixed-window-size samples for modeling makes it harder to identify landslide hazards of different scales, which limits the identification accuracy to some extent [47]. So, a multi-window identification model can perform better in regional potential landslide identification and avoid subjective prejudice of window size relying on expert experience [47]. Further improvements can be made by developing models for identifying multiple scales areas using multiple windows. It is expected that more accurate identification models for delineating anomalous deformation areas could be proposed by improving the above limitations.

## 6. Conclusions

This study provides insights for delineating anomalous InSAR deformation areas automatically through the InSAR technique, hot spot analysis, and a deep learning semantic segmentation model. Our principal conclusions are as follows:

(1) The hot spot map is similar to delineating anomalous deformation areas manually, and can more easily be understood by algorithms, leading to a better guiding effect when extracting anomalous areas than the InSAR deformation map.

(2) The introduction of topographic information can constrain the boundaries of anomalous deformation areas based on the local landform. Among the different combinations of topographic information, introducing aspect as an input feature contributes to the optimal semantic segmentation model.

(3) The U-Net model is more suitable for extracting the boundaries of anomalous areas due to special network structure compared with the SegNet model.

(4) The optimal model was built by combining Google Earth images, the InSAR deformation hot spot map, and aspect based on the U-Net semantic segmentation deep learning algorithm, with the precision, recall, F1 score, and IoU being 0.822, 0.835, 0.823, and 0.705, respectively.

The application of the proposed model enables researchers to reduce the limitations regarding time and labor costs in the process of using InSAR deformation results. Our result can provide insights for delineating potential landslides automatically and provide valuable information for landslide risk mitigation in susceptible areas worldwide.

**Author Contributions:** Conceptualization, Y.L. (Yiwen Liang); methodology, Y.L. (Yiwen Liang) and Y.Z.; validation, Y.L. (Yiwen Liang) and Y.L. (Yuanxi Li); formal analysis, Y.L. (Yiwen Liang); investigation, Y.L. (Yiwen Liang) and Y.Z.; resources, Y.Z.; data curation, Y.L. (Yiwen Liang) and J.X.; writing—original draft preparation, Y.L. (Yiwen Liang) and Y.L. (Yuanxi Li); writing—review and editing, Y.L. (Yiwen Liang) and Y.Z.; visualization, Y.L. (Yiwen Liang) and Y.L. (Yuanxi Li);



supervision, Y.Z.; project administration, Y.Z.; funding acquisition, Y.Z. All authors have read and agreed to the published version of the manuscript.

**Funding:** This research was funded by the Important Talent Project of Gansu Province (Grant No. 2022RCXM033), the Fundamental Research Funds for the Central Universities (Grant No. lzujbky-2021-ey05), National Natural Science Foundation of China (Grant No. 42007232), the Science and Technology Project of Gansu Province (Grant No. 22ZD6FA051), and Research on 3D Geological Modeling and Application Technology for Urban Geological Survey (Grant No. YJKJ-2022-04).

**Data Availability Statement:** Not applicable.

**Acknowledgments:** We are grateful to the editors for helping to significantly improve the quality of the manuscript.

**Conflicts of Interest:** The authors declare no conflict of interest.

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
