# Peer review of "Automatic Identification for the Boundaries of InSAR Anomalous Deformation Areas Based on Semantic Segmentation Model"

_remotesensing, doi:10.3390/rs15215262_

Round 1

Reviewer 1 Report (Previous Reviewer 1)

Comments and Suggestions for Authors

I have no comment.

Author Response

Dear Reviewer,

Thank you very much for your hard work in processing our manuscript entitled “Automatic Identification for the Boundaries of InSAR Anomalous Deformation Areas Based on Semantic Segmentation Model”. We are very grateful for the reviewer's affirmation of our work.

Best regards.

Reviewer 2 Report (Previous Reviewer 3)

Comments and Suggestions for Authors

Thanks for improving the manuscript. I think there is a critical problem that should be carefully addressed.

The authors replied that the deformation information was converted into raster format with values of -234 to 366 rather than 0 to 255. And they didn't reply how many channels are used for the deformation information. In general, the RGB raster dataset has three channels with values of 0-255 for the deep learning network. If the authors use the value of -234 to 366, how to balance the value discrepancy among different channels? For example, the channel of deformation with a value of -234 to 366, the channel of hot spot map with a value from -38835.9 to 17832.8, and the channel of Google Earth image with a value of 0 to 255 for each channel. Balancing or weighting different channels is a critical issue for this study. If all the channels have values of 0-255, the deformation information will be equally the same as other channels, i.e., Google Earth and hot spot map. In addition, please also give the value range of each channel that is used for deep learning.

Author Response

Reviewer 3 Report (New Reviewer)

Comments and Suggestions for Authors

The objective of this paper is to study the automatic identification for the boundaries of InSAR anomalous deformation areas based on semantic segmentation model. In particular, a combination of InSAR techniques, hot spot analysis, and semantic segmentation model is implemented to establish identification models, while a comparison of different input features and different semantic segmentation models is additionally performed. Then, the optimal model is established, and its further directions are concluded.

This is an interesting and well-structured paper. All necessary sections (Introduction, Study area, Datasets and Methodology, Experiment and Results, Discussion, Conclusion) have been considered. Moreover, the “Datasets and Methodology”, “Experiment and Results” and “Discussion” sections are divided into sub-sections, providing additional details. Furthermore, all Figures, Tables and Diagrams are consistent with the analysis provided in the manuscript. However, some changes should be implemented, which will improve the paper. In particular:

Line 13-31: Although the abstract has been properly structured, unnecessary details are contained (these details could be placed in the manuscript). The abstract should be clear and concise, while the most significant processes/findings/conclusions should be highlighted. Please, modify the abstract by reducing its length.

Lines 40-43: This part of the “Introduction” section is not bibliographically supported by the corresponding references. Typical papers, in which the corresponding information can be obtained and optionally be cited, are the following: 1. Kalantar, B., Ueda, N., Saeidi, V., Ahmadi, K., Halin, A. A., & Shabani, F. (2020). Landslide Susceptibility Mapping: Machine and Ensemble Learning Based on Remote Sensing Big Data. Remote Sensing, 12(11), 1737. https://doi.org/10.3390/rs12111737, 2. Karagianni, A., Lazos, I., & Chatzipetros, A. (2019). Remote Sensing Techniques in Disaster Management: Amynteon Mine Landslides, Greece. In Lecture Notes in Geoinformation and Cartography. https://doi.org/10.1007/978-3-030-05330-7_9, 3. Liu, P., Wei, Y., Wang, Q., Chen, Y., & Xie, J. (2020). Research on Post-Earthquake Landslide Extraction Algorithm Based on Improved U-Net Model. Remote Sensing, 12(5), 894. https://doi.org/10.3390/rs12050894. Please, apply.

Lines 128-130: Please, add a map and/or a lithostratigraphic column with the major lithological formation, described in the manuscript.

Line 157: Please, provide a more detailed description in the Figure 2 caption.

Line 217: Please, provide Figure 3 in a higher resolution. It contains blurry parts in the current form.

Line 416: Similarly, provide a more detailed description in the Figure 10 caption, please.

Line 663: The “Conclusions” section should be modified. In its current form it resembles an abstract rather than conclusion. This section should be comprehensive, while the major findings of the paper should be highlighted. Maybe, numbering of the concluding remarks could be performed. Please, apply.

Round 2

Reviewer 2 Report (Previous Reviewer 3)

Comments and Suggestions for Authors

Thanks for the improvement. No more comments.

This manuscript is a resubmission of an earlier submission. The following is a list of the peer review reports and author responses from that submission.

Round 1

Reviewer 1 Report

Comments and Suggestions for Authors

The manuscript proposes a method of using deep learning semantic segmentation model combined with Google Earth image, InSAR deformation hot spot map, and topographic information to identify the boundaries of anomalous deformation areas automatically, which improves the automation of image interpretation to a certain extent, and seems to be a valuable idea. However there are the following problems that need to be solved:
(1) What is the value of deformation region boundary extraction for landslide identification?

(2) How to guarantee the accuracy of boundary annotation when sample annotation, and what features are relied on for annotating?

(3) Although the range of anomalous regions is less than 256 pixels, there are also anomalous regions that are segmented into different pictures after segmentation, how to solve this problem?

(4) The manuscript describes slope, direction and elevation as good supplementary features, slope and direction have good consistency in small areas of the slope and can be considered as effective features with differentiation ability, but elevation does not have consistency in the slope, does the feature have differentiation ability? Did the authors try to utilize different combinations of topographic information to test the distinguishing ability of each information?

(5) The results in Figure 11 do not seem to have a high degree of agreement, and it is suggested to utilize more scientific indexes to evaluate its accuracy, such as Intersection of Union (IoU).

Comments on the Quality of English Language

No comments.

Reviewer 2 Report

Comments and Suggestions for Authors

Please see the review comments.

Comments on the Quality of English Language

Please see the review comments.

Reviewer 3 Report

Comments and Suggestions for Authors

Here are my comments to improve the manuscript.

1. How was the satellite image from Google Earth pre-processed? The image tiles from Google Earth may have different brightness, tones, and distortions. The image preprocessing steps should be explained in more detail.

2. In section 3, there lacks a part to introduce the data preprocessing step, which is essential to this study. The image resolution of the InSAR, Google Earth image, and DEM is different. Was the topographic data unsampled to the same as the Google Earth image?

3. In section 4.1.1, the authors mentioned that “In the experiment, InSAR deformation result was used as the input feature by converting the coherent targets to raster dataset in ArcGIS”. I reckon that the InSAR deformation velocity is converted to a channel to be input to the model with Google Earth image, as shown in the first column of Figure 11. How do you convert the velocity to three RGB channels?

4. I don’t understand why not use the 12.5 m resolution ALOS DEM (used as the topographic dataset in Figure 2) to remove the topographic phase during IPTA processing?

5. In section 3.3, the authors introduce the wij as the spatial weight but give no explanation or formation about it. The hot spot map is calculated according to the density of CTs. However, sparse CTs with large deformation velocities should be evaluated as anomalous rather than dense CTs with small deformation velocities. How was this issue addressed?

6. In Figure 7, what is the unit for the kernel density? Quantities indicator should be given, rather than high and low. According to Figure 6, is the kernel density of the areas with no CTs nearly zero? If it is, why does density have signs of both positive and negative, as shown in the color scale of Figure 7?

7. In section 4.1.4, the authors mentioned that “The boundaries of the anomalous deformation areas were determined based on the difference of texture, color, brightness, …… the boundaries of the deformation areas were interpreted accurately.” What is the scheme to determine the boundaries with the texture, color, brightness, and vegetation cover of the Google Earth image and even field surveys? In subfigure (c)-1 of Figure 8, the boundary is obviously not delineated according to the brightness.

8. InSAR deformation velocity is derived for more than a year. However, the level 16 Google Earth image was captured on March 11, 2021, which means the boundaries determined from the Google Earth image are always overestimated. Does this influence the results?

9. I cannot believe the first model reaches only a precision of 0.2, which is a large difference from others. There might be some problems during the data processing or training. Adding topographic information gives a minor improvement in the results and complexing the data processing and training. There is no point in adding this information.

Reviewer 4 Report

Comments and Suggestions for Authors

In this paper, the authors try to identify the boundaries of anomalous deformation areas automatically using deep learning semantic segmentation method. This research is helpful for landslide hazard mitigation and risk management. However, some problems should be addressed before publication. My main comments are as follow:

1) the introduction should be reorganized. the novelty of this paper should be enhanced. moreover, some latest researches should be cited in the introduction, such as "Wu, Z., Ma, P., Zheng, Y., Gu, F., Liu, L., & Lin, H. (2023). Automatic detection and classification of land subsidence in deltaic metropolitan areas using distributed scatterer InSAR and Oriented R-CNN. Remote Sensing of Environment, 290, 113545.";

2)  the other dataset used in this paper should be added and described.

3)In lines 322-323, "After the extensive field surveys, the boundaries of the deformation areas were interpreted  accurately. ", it is suggested to give the field photos in the paper.

4) section 4.4 should be moved to the method section.

5) The discussion part is superficial. more discussion can be conducted, such as the effects of the parameters of the model on the results. 

Comments on the Quality of English Language

The English should be checked and improved.

Round 2

Reviewer 2 Report

Comments and Suggestions for Authors

The figure quality of Figure 3 is not high.

Comments on the Quality of English Language

None.

Reviewer 3 Report

Comments and Suggestions for Authors

Thanks for improving the manuscript. However, some of my comments have not yet been addressed. Please review the comments carefully and replay them in detail.

  1. The deformation velocities are point-wise results in float numbers. Did you convert it into 0-255? Of one channel? Were these point-wise results converted to raster format? For the areas with no coherent points, what are the pixel values in the converted channel?
  2. I don’t think ALOS DEM performs worse than 1-arc-second SRTM DEM in phase unwrapping for all cases. The resolution of ALOS DEM is better than SRTM DEM. It is better to demonstrate the issue of phase unwrapping in this case using both DEMs.
  3. In Figure 7, what is the kernel density value of green colors? According to the equation (2), a negative value is impossible for the calculated density. Kernel density may have no physical unit, but the authors should explain what the two ends of the kernel density spectrum stand for.
  4. In response by the authors, the boundaries of anomalous deformation were determined by landslide experts. It is too subjective and difficult to repeat the proposed framework for getting the same results. If the boundaries are different for each expert, the results will be largely biased.
